# Unraveling the *IDH3A*: Expanding the Genotypic Spectrum of Macular Pseudocoloboma

**DOI:** 10.3390/ijms26073364

**Published:** 2025-04-03

**Authors:** Mirjana Bjeloš, Ana Ćurić, Benedict Rak, Biljana Kuzmanović Elabjer, Mladen Bušić, Katja Rončević

**Affiliations:** 1University Eye Department, Reference Center of the Ministry of Health of the Republic of Croatia for Pediatric Ophthalmology and Strabismus, Reference Center of the Ministry of Health of the Republic of Croatia for Inherited Retinal Dystrophies, Reference Center of the Ministry of Health of the Republic of Croatia for Standardized Echography in Ophthalmology, University Hospital “Sveti Duh”, Sveti Duh 64, 10000 Zagreb, Croatia; dr.mbjelos@gmail.com (M.B.); benedict.rak@gmail.com (B.R.); belabjer@kbsd.hr (B.K.E.); mbusic@kbsd.hr (M.B.); 2Faculty of Medicine, Josip Juraj Strossmayer University of Osijek, Josipa Huttlera 4, 31000 Osijek, Croatia; 3Faculty of Dental Medicine and Health Osijek, Josip Juraj Strossmayer University of Osijek, Crkvena 21, 31000 Osijek, Croatia; 4School of Medicine of the Catholic University of Croatia, Ilica 244, 10000 Zagreb, Croatia; katja.roncevic@gmail.com

**Keywords:** coloboma, retinitis pigmentosa, atrophy, genotype, phenotype

## Abstract

Disease-causing variants in the *IDH3A* gene are associated with autosomal recessive retinitis pigmentosa 90 (RP90) and Leber congenital amaurosis, with or without macular pseudocoloboma. Here, we report two patients: one compound heterozygous for *IDH3A* c.364G>A, p.(Ala122Thr), which has conflicting classifications, and for *IDH3A* c.293C>T, p.(Pro98Leu), which is likely pathogenic, and the other homozygous for *IDH3A* c.364G>A, p.(Ala122Thr). This study is aimed at providing evidence to link the latter variants to a clinical phenotype. The first patient was a 6-year-old girl, and the second patient was a 29-year-old female. In both patients, the diagnostic assessments were consistent with the phenotype of RP, characterized by macular pseudocoloboma but of varying severity. Patients’ phenotypes suggest that the c.293C>T, p.(Pro98Leu) variant is linked to a more severe and extensive clinical phenotype, while the c.364G>A, p.(Ala122Thr) variant results in a milder condition, primarily limited to retinal involvement. In Patient 2, the presence of both global and local stereopsis, indicating advanced visual development, suggests that the p.(Ala122Thr) missense variant may act as a hypomorphic allele which likely allows for some residual enzymatic activity of the IDH3A protein. This report highlights that macular pseudocoloboma can manifest even in the absence of a null variant. In contrast, more severe symptoms, such as mitochondrial encephalopathy, seem to be associated with the presence of a null allele. Furthermore, to the best of our knowledge, this is the first report of the *IDH3A* c.293C>T, p.(Pro98Leu) variant.

## 1. Introduction

In mammals, isocitrate dehydrogenase 3 (IDH3) is a heterotetramer composed of two alpha, one beta, and one gamma subunit [1]. The *IDH3A* gene (*MIM 601149*) encodes the catalytic alpha subunit, a 339 amino acid long protein, which, along with beta (IDH3B) and gamma (IDH3G) subunits, drives the oxidative decarboxylation of isocitrate to α-ketoglutarate in the tricarboxylic acid (TCA) cycle. This reaction generates NADH, which fuels ATP production in mitochondria [2]. Autosomal-recessive mutations in the enzymes of the TCA cycle have been extensively documented, often in association with severe neurological deficits [1]. However, while mutations in IDH are exceedingly rare, those affecting the IDH3B subunit have been reported only in patients with retinitis pigmentosa carrying homozygous loss-of-function variants [3]. The first report of a recessive mutation in the *IDH3A* gene was published in 2017 [1]. Similarly to other recessive TCA cycle defects, this mutation was linked to a severe neurological phenotype, primarily involving the brainstem and basal ganglia [1]. However, the *IDH3A* mutation appears to exert a detrimental effect from early fetal development, as evidenced by the occurrence of intrauterine growth retardation [1]. This early disruption progresses to early-onset dysfunction, affecting not only the central nervous system but also the peripheral and autonomic nervous systems postnatally [1].

Recent findings indicate that disease-causing variants in the *IDH3A* gene are increasingly associated with autosomal recessive retinitis pigmentosa 90 (RP90) and Leber congenital amaurosis (LCA), with or without macular pseudocoloboma. The HGMD Professional variant database (version 2021.1) has cataloged nine disease-causing variants, the majority of which are missense variants (78%) [4].

Here, we report two patients: one compound heterozygous for *IDH3A* c.364G>A, p.(Ala122Thr), which has conflicting classifications, and for *IDH3A* c.293C>T, p.(Pro98Leu), which is likely pathogenic, and the other homozygous for *IDH3A* c.364G>A, p.(Ala122Thr). To the best of our knowledge, this is the first report of the *IDH3A* c.293C>T, p.(Pro98Leu) variant. This study is aimed at providing evidence to link the latter variants to a clinical phenotype as both patients manifested RP with macular pseudocoloboma but of varying severity.

## 2. Case Description

### 2.1. Patient 1

The first case was a 6-year-old girl referred to our Reference Center for comprehensive clinical evaluation and genetic analysis. The patient’s nystagmus was noticed from the 4th month of life, and later poor orientation in low light conditions was present. At the first ophthalmological examination at the age of 6 months, suspicion of Best’s disease was raised. At the age of 3 years, she was hospitalized at the Clinic for Infectious Diseases under the diagnosis of acute cerebelitis (the causative agent has not been isolated). After that, psychomotor development decelerated, consistent with infantile encephalopathy.

At clinical examination, the impression of photophobia was evident. There was a constant, pendular, horizontal low amplitude and medium frequency nystagmus. Her best-corrected visual acuity (BCVA) assessed using the Lea Symbols inline chart at 3 m measured 1.0 logMAR tested binocularly, as well as monocularly. At a testing distance of 40 cm, the BCVA was 0.7 logMAR binocularly, and 0.8 logMAR monocularly. Lang and Titmus tests were defective.

Farnsworth’s D-15 dichotomus test and Lanthony desaturated 15-hue panel revealed pathologic color discrimination of the diffuse type. The CSV-1000 contrast sensitivity test for the spatial frequencies of 3, 6, 12, and 18 cpd measured 1.49 log units, 1.55 log units, 0.91 log units, and 0.17 log units when tested binocularly, 1.49 log units, 1.38 log units, 1.08 log units, and 0.17 log units when tested with the right eye (RE), and 1.63 log units, 1.21 log units, 1.08 log units, and 0.47 log units when tested with the left eye (LE).

Microperimetry and automated kinetic perimetry could not be performed due to unstable fixation.

Pattern reversal visual evoked potential (p-VEP) (Roland Consult RETIport/scan 21, Roland Consult Stasche and Finger GmbH–German Engineering, Brandenburg an der Havel, Germany), which was recorded using goldcup electrodes according to the International Society for Clinical Electrophysiology of Vision (ISCEV) standards [5], demonstrated a bilaterally barely maintained waveform morphology with reduced amplitude and normal implicit P100 time.

Optos^®^ California (Optos California, Dunfermline, UK) ultra-widefield imaging depicted bilateral well-defined atrophic lesions in the macula measuring 4 × 2.9 mm on the RE, and 4.4 × 3.2 mm on the LE. Optic nerve heads (ONH) were grayish and narrowing of the vasculature was evident with the almost avascular periphery. Throughout the middle periphery towards the far periphery, whitish dot deposits could be seen, only a few bone spicules, and dark pigmented punctiform deposits (Figure 1).

Fundus autofluorescence (FAF) revealed a significant reduction in autofluorescence in the area of pseudocoloboma. Fluorescence further diffusely increased towards the arcades, and further decreased towards the periphery (Figure 2).

HRA + OCT Spectralis^®^ (Heidelberg Engineering, Heidelberg, Germany) optical coherence tomography (OCT) imaging displayed bilateral pseudocoloboma of the macula with significant atrophy of the neurosensory retina, retinal pigment epithelium (RPE), and choriocapillaris, with the persistence of large choroidal vessels (Figure 3).

Retinoscopy revealed a refractive error of +1.50 Dsph/−1.75 Dcyl ax 10° on the RE, and +2.25 Dsph/−2.25 Dcyl ax 0° on the LE.

A saliva sample was obtained for genetic testing, and sequencing using the Blueprint Genetics Retinal Dystrophy Panel Plus (version 7, 30 October 2021) demonstrated compound heterozygosity for *IDH3A* c.364G>A, p.(Ala122Thr), and *IDH3A* c.293C>T, p.(Pro98Leu). Next-generation sequencing data strongly suggested that these variants are in trans, which likely explains the patient’s clinical presentation.

### 2.2. Patient 2

The second patient was a 29-year-old female presenting with nyctalopia. Since early childhood, she had been unable to navigate in low-light conditions. At the time of her primary school admission, her vision was unremarkable; however, by age eight, the visual acuity on BE declined to 0.3 logMAR. There are no other visually impaired members in the family. She had no neurological signs and symptoms.

Upon her clinical examination in our reference center, her BCVA assessed with ETDRS inline chart at 3 m was 0.8 logMAR tested binocularly, and monocularly on the RE, while that on the LE measured 0.6 logMAR. At 40 cm, the BCVA measured 0.7 logMAR binocularly, and monocularly. On a Lang II stereotest, she was able to identify the star and the moon, and the Titmus fly test was 100 s of arc.

Color vision testing revealed a tritanomaly on the Farnsworth D-15 dichotomous test. The Lanthony desaturated 15-hue panel showed a diffuse pattern of pathological color discrimination, evident in both monocular and binocular conditions. The CSV-1000 contrast sensitivity test for the spatial frequencies of 3, 6, 12, and 18 cpd was reduced to 1.78 log units, 1.55 log units, 1.08 log units, and 0.81 log units when tested binocularly, 1.78 log units, 2.14 log units, 1.08 log units, and 0.64 log units when tested with the RE, and 1.78 log units, 1.99 log units, 1.25 log units, and 0.64 log units when tested with the LE.

MAIA microperimetry (iCare Finland Oy, Vantaa, Finland) (20°, 68 points) recorded reduced mean retinal sensitivity with a threshold of 9.2 dB on the RE and 8.8 dB on the LE with unstable fixation in both eyes (BE). Goldmann perimetry III4e stimulus identified a sum of meridians of 728° on the RE and 734° on the LE. Octopus^®^ (Octopus 900 perimetry, Haag Streit International, Koeniz, Switzerland) G, TOP, SAP, *w*/*w*, III, and 10-2 dynamic, static perimetry (false positives 0%, false negatives 0%) demonstrated diffuse retinal sensitivity loss with a mean sensitivity of 18.0 dB on the RE and 15.6 dB on the LE.

Biomicroscpy revealed mild posterior subcapsular cataracts on BE. HRA + OCT Spectralis^®^ imaging depicted bilateral pseudocoloboma of the macula with RPE and choriocapillaris atrophy, and the disruption of the outer retinal layers. Compared to Patient 1, the atrophy was less severe (Figure 4).

Optos^®^ California ultra-widefield imaging depicted a pale ONH surrounded by a wide peripapillary atrophy, with the attenuated retinal vasculature not reaching the periphery, and abundant pigment accumulations in the bone spicule type. Macular pseudocoloboma was present bilaterally, characterized by well-demarcated atrophic lesions, measuring 1.5 × 1.7 mm (RE) and 1.6 × 1.4 mm (LE), which were milder in appearance than those in Patient 1 (Figure 5).

FAF revealed significant hypoautofluorescence corresponding to the pseudocoloboma lesions (Figure 6).

p-VEP performed using goldcup electrodes according to the ISCEV standards [5], revealed reduced N75-P100 amplitudes, along with variable implicit P100 times. Full-field electroretinography (FFERG), performed following ISCEV standards [6], revealed an almost flattened response curve under both scotopic and photopic conditions.

A saliva sample was obtained for genetic testing and sequencing using the Blueprint Genetics Retinal Dystrophy Panel Plus (version 7, 30 October 2021) and demonstrated homozygosity for *IDH3A* c.364G>A, p.(Ala122Thr).

In both Patient 1 and Patient 2, the clinical and diagnostic findings were consistent with the phenotype of retinitis pigmentosa associated with macular pseudocoloboma.

## 3. Discussion

### 3.1. IDH3A

The total ablation of IDH3A is not compatible with life [7]. The IDH3A subunit serves as the catalytic component of the IDH3 complex, directly facilitating the conversion of isocitrate to α-ketoglutarate, a crucial step in the TCA cycle. Damage or dysfunction of the alpha subunit results in the complete loss of the catalytic activity of the entire enzyme complex, thereby disrupting the production of NADH. This disruption significantly impairs the TCA cycle and cellular energy production. In the absence of NADH generation, the electron transport chain cannot function effectively, leading to a severe energy deficit within the cell. This energy imbalance can have profound consequences on cellular metabolism, particularly in high-energy-demand tissues such as the heart, brain, liver, kidneys, and muscles.

In the AlphaFold model of IDH3A, the overall fold is a two-domain structure that is typical of IDH family enzymes, with a large central β-sheet flanked by α-helices. The active site which binds isocitrate and NAD⁺, is in a cleft between the domains. IDH3A subunit interfaces (with IDH3B and IDH3G) are mediated by specific regions called “clasp domains”. In the structure, residue Pro98 lies at the end of a short β-strand within the central β-sheet, while Ala122 is situated in a loop connecting this sheet to an adjacent helical region. Neither residue is directly in the active site; however, their structural positions suggest that they contribute to maintaining the enzyme’s architecture: Pro98 appears to introduce a sharp turn in the β-sheet, and Ala122 resides in a flexible loop near a subunit interface (the “clasp” region) (Figure 7) [8].

While damage to IDH3B or IDH3G would also compromise the function of the IDH3 complex, their effects would be less detrimental than that of the alpha subunit. IDH3B and IDH3G primarily contribute to structural support and regulatory functions within the complex. Dysfunction in these subunits is likely to impair the assembly or stability of the enzyme complex, potentially reducing its efficiency. However, some residual enzymatic activity may persist if the alpha subunit is partially functional. The overall impact of damage to these subunits is contingent upon the extent of the dysfunction. Impaired assembly could lead to a reduction in IDH3 complex activity, but a complete loss of function, as seen with damage to IDH3A, would be less likely. This distinction may elucidate the lack of reported disease associations for the gene encoding IDH3G [9], whereas variants in IDH3B are typically associated with the absence of macular pseudocoloboma or systemic involvement [9]. In contrast, specific variants of *IDH3A* have been implicated in macular pseudocoloboma and severe infantile encephalopathy, conditions affecting tissues such as the macula and brain, which are highly metabolically active and, therefore, particularly vulnerable to energy deficits [9].

Pierrache et al. identified biallelic *IDH3A* variants in seven patients from four families [9]. All patients had early-onset night blindness and were diagnosed between the ages of 1 and 11 years. Three patients had bilateral early-onset macular pseudocoloboma. Interestingly, all patients with macular pseudocoloboma were compound heterozygotes, carrying one null variant, and exhibited a more severe phenotype [9]. Further, *IDH3A* variants have been reported in two additional cases, one affected with LCA accompanied by macular pseudocoloboma [10], and the other with autosomal recessive RP with dark adaptation problems since the age of 5 years, a diagnosis of RP at 28 years, and severe peripheral and central retinal degeneration including the posterior pole at 46 years [7]. In addition, a homozygous missense variant, *IDH3A* c.911C>A, p.(Pro304His), has been identified through exome sequencing in a patient with severe infantile encephalopathy and RP [1].

### 3.2. IDH3A c.364G>A, p.(Ala122Thr)

The variant is located in the fifth exon of *IDH3A*, a region that has previously been associated with other pathogenic variants [7].

It is also reported at a very low frequency in population databases, with an allele frequency of 0.0009% in gnomAD [11]. There is one individual that is heterozygous for this variant in gnomAD. All in silico tools utilized (Polyphen, SIFT, MutationTaster) predict this variant to be damaging to protein structure and function. *IDH3A* c.364G>A, p.(Ala122Thr) has been reported in a homozygous state in a 54-year-old male from Bosnia and Herzegovina without a presence of pseudocoloboma [7]. The patient was reported to have four siblings, all with normal vision and no history of retinal degeneration. The *IDH3A* c.364G>A, p.(Ala122Thr) variant was submitted to ClinVar (variation ID 977474), and there has conflicting classifications of pathogenicity: pathogenic, and uncertain significance [12]. This particular amino acid residue is also found in the two human paralogous IDH3 subunits, IDH3B and IDH3G, which have a related structure and share over 30% sequence identity.

Alanine at position 122 lies within a flexible loop (residues ~120–126) that connects a β-strand (residues 114–119) to a nearby α-helix [8]. In the wild-type structure, Ala122 is a small, nonpolar residue that is buried within the loop, helping to maintain its mobility and avoiding steric interference with surrounding backbone atoms. The p.Ala122Thr variant introduces a larger, polar threonine side chain at this position. Structural modeling predicts that the new side chain can form a hydrogen bond with a nearby backbone atom, inducing a more rigid structural environment. This substitution may perturb loop dynamics that are important for IDH3 function. Increased rigidity at this site could reduce conformational flexibility, impair allosteric movements, or disrupt inter-subunit contacts. Furthermore, the introduction of a polar residue into a predominantly hydrophobic pocket may be energetically unfavorable, potentially decreasing protein stability. In a multi-subunit enzyme like IDH3, such local disruptions could hinder assembly or reduce the catalytic efficiency of the heterotetrameric complex (Figure 7) [8]. The p.Ala122Thr variant is therefore predicted to alter loop mobility and contribute to reduced enzymatic activity, supporting its classification as functionally impactful in disease contexts such as retinitis pigmentosa.

### 3.3. IDH3A c.293C>T, p.(Pro98Leu)

This variant which is absent in gnomAD is predicted to be deleterious by all in silico tools utilized (PolyPhen-2 version 2.2.3, SIFT version 6.2.1, MutationTaster 2025). The affected amino acid is highly conserved across mammals and other vertebrates, which suggests that this position may not tolerate variation. To our knowledge, this variant has not previously been documented in the peer-reviewed literature or listed in clinically relevant variation databases. Based on the established association between the gene and the patient’s phenotype, the variant’s rarity in control populations, in silico predicted pathogenicity, and in-*trans* occurrence with a pathogenic variant, *IDH3A* c.293C>T, p.(Pro98Leu) was classified as likely pathogenic [13].

Proline at position 98 is located at a β-turn between two β-strands within the central β-sheet of IDH3A. Its cyclic structure introduces a rigid kink in the backbone that helps stabilize the turn and properly align adjacent secondary structure elements. Substitution with leucine removes this conformational constraint, introducing flexibility that may alter the turn geometry and misalign the following β-strand. Structural modeling predicts a slightly more extended conformation in the mutant, with the leucine side chain protruding into space previously occupied by proline’s ring. This may cause subtle steric clashes or local loop deformation. Given the role of this region in β-sheet stability and overall fold integrity, the p.Pro98Leu variant is predicted to locally destabilize the IDH3A structure. In a multi-subunit enzyme of IDH3, such localized misfolding may impair subunit association, reduce protein stability, or hinder the assembly of the catalytically active heterotetramer (Figure 7) [8]. This is supported by the variant’s association with retinitis pigmentosa in affected individuals, consistent with a loss-of-function mechanism.

### 3.4. Macular Pseudocoloboma

Macular pseudocoloboma was described first by Clausen in 1921 in a family with a dominant inheritance pattern IRD and no systemic involvement [14]. They were also mentioned in Foxman’s classification of congenital and early-onset RP as a common finding in patients with uncomplicated (or nonsyndromic) LCA [15].

Genes associated with macular pseudocoloboma are involved in different pathways, such as energy metabolism (*IDH3A*, *NMNAT1*), pre-mRNA splicing (*DHX38*), transcription control and transcription factors (*PRDM13*, *CRX*), and extracellular matrix proteins (*COL18A1*). However, the disease manifestation seems to be restricted to the retina [6]. Among them, mutations in *NMNAT1* are considered to be the most commonly accompanied by macular pseudocoloboma [16]. There are no reports on IRD patients with two null variants in *IDH3A*, *NMNAT1*, and *DHX38*, leading to the assumption that the complete inactivity of these proteins may be incompatible with life [9].

While true colobomas are the result of the failed or incomplete closure of the embryonic fissure during the sixth and seventh week of fetal development and are often associated with microphthalmos or other closure defects, the term “macular coloboma” is often misused for describing atrophic lesions in the central retina [9]. The fundus characteristic of true macular coloboma is typically distinguished by a well-defined, punched-out atrophic lesion measuring approximately 3 to 6 disk diameters in the central macula [17]. The lesion is associated with the atrophy of the neurosensory retina, with or without the persistence of large choroidal vessels, and, as a requisite, the absence of both the choriocapillaris and RPE [18].

Previous reports describing variants in the *IDH3A* gene across various families have consistently noted a relatively severe phenotype, with rapid progression from night blindness to complete vision loss [9,10]. In some cases, these variants have been associated with early-onset, severe retinal dystrophy, accompanied by typical macular pseudocoloboma [10]. Notably, while some patients presented with macular pseudocoloboma, this feature was not observed in all cases [9,10]. In a cohort reported by Pierrache et al., patients were found to carry either two missense variants or a combination of a missense and a null variant [9]. The authors suggested that certain missense variants might act as hypomorphic variants, permitting residual IDH3A activity [9]. Notably, all patients with autosomal recessive RP and macular pseudocoloboma carried a null variant in compound heterozygosity, which could account for the more severe phenotype observed in these individuals. However, an interesting case involved a sibling with only mild foveal abnormalities, who carried the same combination of null and missense variants as her brother, who exhibited macular pseudocoloboma. The authors hypothesized that the phenotypic difference between the siblings could be attributed to variations in *IDH3A* expression. A higher expression of the mutant IDH3A mRNA, carrying the missense variant, in the sibling with the milder phenotype may have contributed to this difference. Nevertheless, considering the strict stochiometric balance required for the NAD-IDH protein complex, the authors speculated that differential expression levels of IDH3B and IDH3G could also influence the resulting phenotypic outcome [9].

The fundus alterations observed in our patients are consistent with macular pseudocoloboma; however, the size of the lesions does not appear to correlate with the degree of visual acuity impairment. The severity of the clinical presentation is more closely associated with the specific nature of the genetic variant involved. Notably, the phenotypic differences between Patient 1 and Patient 2 suggest that the c.293C>T, p.(Pro98Leu) variant is linked to a more severe and extensive clinical phenotype, while the c.364G>A, p.(Ala122Thr) variant results in a milder condition, primarily limited to retinal involvement.

In Patient 2, the presence of both global and local stereopsis, indicating advanced visual development, suggests that the p.(Ala122Thr) missense variant may act as a hypomorphic allele. This variant likely allows for some residual enzymatic activity of the IDH3A protein, which could explain the milder phenotype and more preserved visual development despite the mutation.

Furthermore, we hypothesize that the infantile encephalopathy, along with cognitive and motor decline observed in Patient 1 may be attributed to a mitochondrial encephalopathy caused by mutations in the *IDH3A* gene. Diagnosing this condition was challenging due to the absence of specific biomarkers to support this hypothesis, as genetic analysis had not been performed at the time. Considering that Patient 1 is heterozygous for both the p.(Ala122Thr) and p.(Pro98Leu) variants, we propose that the *IDH3A* c.293C>T, p.(Pro98Leu) variant may represent a null mutation, contributing to the more severe symptoms observed in this patient.

## 4. Conclusions

This report highlights that macular pseudocoloboma can manifest even in the absence of a null variant. In contrast, more severe symptoms, such as mitochondrial encephalopathy, seem to be associated with the presence of a null allele. Furthermore, to the best of our knowledge, this is the first documented clinical presentation linked to *IDH3A* c.293C>T, p.(Pro98Leu).

## Figures and Tables

**Figure 1 ijms-26-03364-f001:**
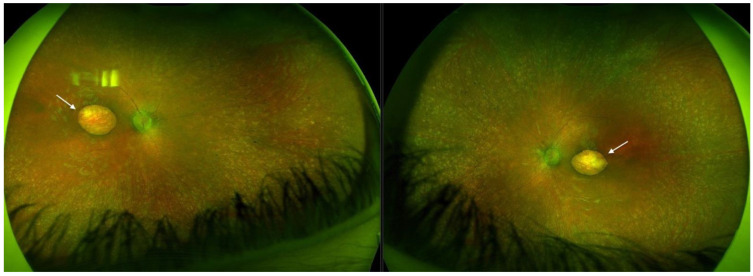
Ultra-widefield imaging in Patient 1 depicted bilateral well-defined atrophic lesions in the macula consistent with macular pseudocoloboma (white arrows, right eye and left eye, respectively).

**Figure 2 ijms-26-03364-f002:**
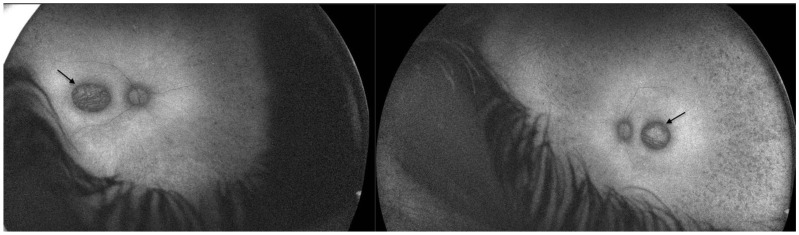
Fundus autofluorescence in Patient 1 revealed a significant reduction in autofluorescence in the area of pseudocoloboma (black arrows, right eye and left eye, respectively).

**Figure 3 ijms-26-03364-f003:**
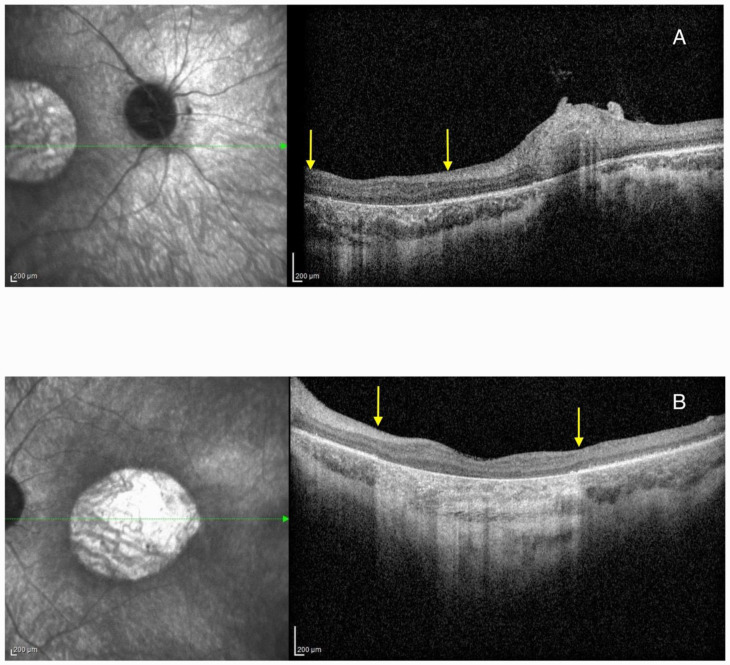
Optical coherence tomography (OCT) imaging showing pseudocoloboma of the macula (area between the yellow arrows) in Patient 1 on the right eye (**A**) and left eye (**B**). The green arrow represents the cross-section through the area of the retina that was presented on the OCT image.

**Figure 4 ijms-26-03364-f004:**
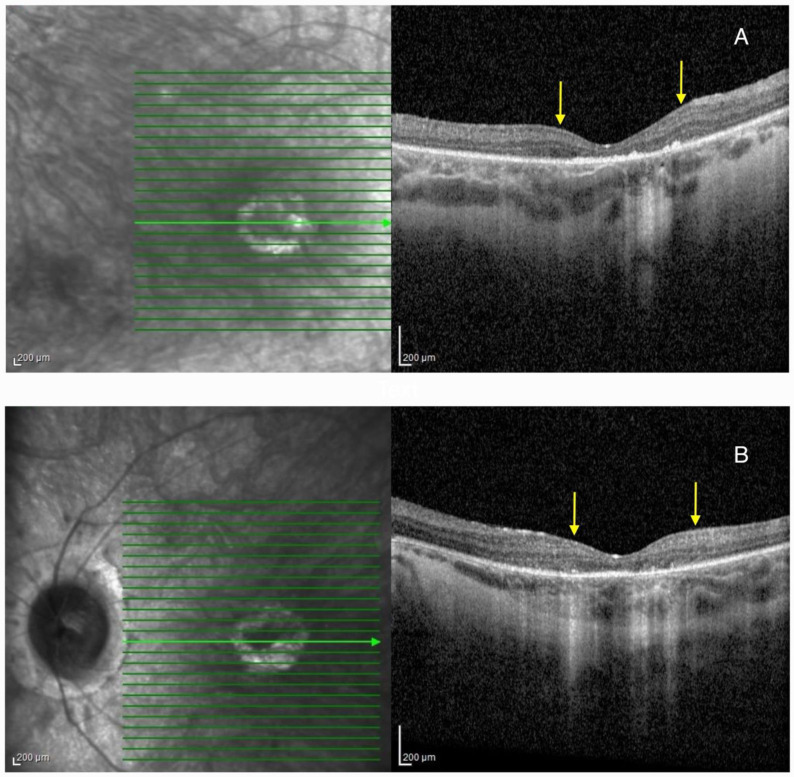
Optical coherence tomography (OCT) imaging showing pseudocoloboma of the macula (area between the yellow arrows) in Patient 2 on the right eye (**A**), and left eye (**B**). Green lines represent cross-sections through the retina, while the green arrow represents the slice that was presented on the OCT image.

**Figure 5 ijms-26-03364-f005:**
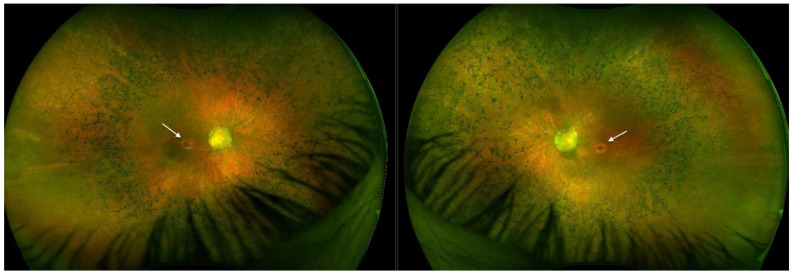
Ultra-widefield imaging in Patient 2 depicted bilateral well-defined atrophic lesions in the macula (white arrows, right eye and left eye, respectively), however they were less noticeable than those in Patient 1.

**Figure 6 ijms-26-03364-f006:**
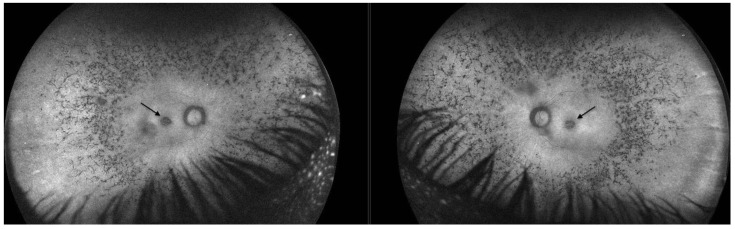
Fundus autofluorescence in Patient 2 revealed a significant reduction in autofluorescence in the ring area of the pseudocoloboma (black arrows, right eye and left eye, respectively).

**Figure 7 ijms-26-03364-f007:**
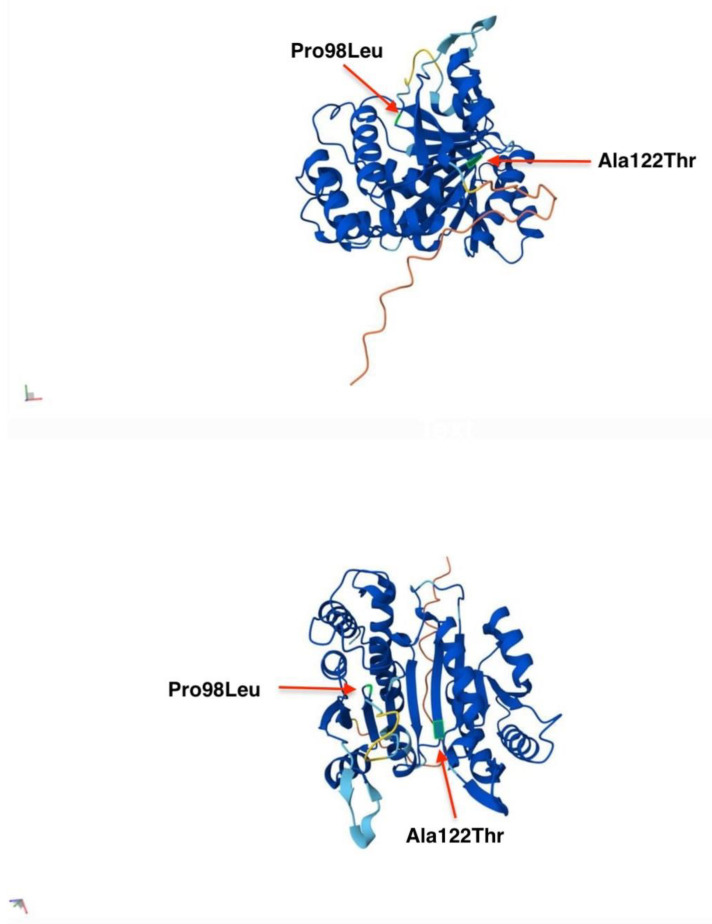
AlphaFold model of IDH3A from different angles of view. Residues Pro and Ala are highlighted in green and indicated with red arrows at their respective positions in the IDH3A protein model.

## Data Availability

The original contributions presented in this study are included in the article. Further inquiries can be directed to the corresponding author.

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
