# Peer review of "Unraveling the IDH3A: Expanding the Genotypic Spectrum of Macular Pseudocoloboma"

_ijms, 2025, doi:10.3390/ijms26073364_

Round 1
Reviewer 1 Report
Comments and Suggestions for Authors
The work of Bjelos et al. reports on a novel disease-causing variant in the gene encoding IDH3A. Mostly, the data presented is of good quality and technically sound. Thus, the manuscript is interesting for the readership of MDPI IJMS.
My concerns:
The title of the suggest that only null variants of the IDH3A gene are causing disease. I'm not sure whether in humans any true null variant of the IDH3A gene has been described so far in respect to macular defects. As stated in the manuscript, null mutation in both alleles are supposed to be lethal. To illustrated this, they cited a paper of Peter et al. in line 180. However, this paper is a case report only, not reporting on the lethality of the complete IDH3A loss-of-function. To my knowledge, the only evidence for a lethal phenotype was shown in knock-out mice (Finlay et al, 2018).
The author should include an AlphaFold based 3D-model of their identified variants to improve their discussion. Of course, experimentally it would be helpful if they could measures remaining enzymatic activity.
The scope of the journal are researchers with a general interest in molecular mechanisms. They are often no MD's and by far no ophthalmologists. Thus, the authors should take this into account and provide a revised version, which is more suited for a more general readership. E.g., provide more information on the identification of a pseudocoloboma in general and should use (more) arrows in the images to areas they discuss (See figure 1, how to identify the "well-defined" atrophic lesion" in the macula consistent with "macular pseudocoloboma").
Author Response
Point-by-point response to reviewer
Dear Reviewer,
the authors thank you for your comments.
We have revised the manuscript entitled “Unraveling the IDH3A gene: macular pseudocoloboma in the absence of a null variant“ (manuscript ID: ijms-3544648) accordingly. The revised parts of the manuscript are highlighted in yellow.
Authors' answers to reviewer’s comments:
Reviewer 1:
The work of Bjelos et al. reports on a novel disease-causing variant in the gene encoding IDH3A. Mostly, the data presented is of good quality and technically sound. Thus, the manuscript is interesting for the readership of MDPI IJMS.
My concerns:
The title of the suggest that only null variants of the IDH3A gene are causing disease. I'm not sure whether in humans any true null variant of the IDH3A gene has been described so far in respect to macular defects. As stated in the manuscript, null mutation in both alleles are supposed to be lethal. To illustrated this, they cited a paper of Peter et al. in line 180. However, this paper is a case report only, not reporting on the lethality of the complete IDH3A loss-of-function. To my knowledge, the only evidence for a lethal phenotype was shown in knock-out mice (Finlay et al, 2018).
Authors’ response: The authors align with the reviewer's assessment. As stated in the manuscript Pierrache et al. identified biallelic IDH3A variants in seven patients from four families. All patients had early-onset night blindness and were diagnosed between the ages of 1 and 11 years. Three patients had bilateral early-onset macular pseudocoloboma. In-terestingly, all patients with macular pseudocoloboma were compound heterozygotes, with one of the variant being a null variant and exibiited a more severe phenotype (lines 275-277). Indeed, there are no reports on IRD patients with 2 null variants in IDH3A, leading to the assumption that complete inactivity of these proteins may be incompatible with life (lines 358-360) [Pierrache, L.H.M.; Kimchi, A.; Ratnapriya, R.; Roberts, L.; Astuti, G.D.N.; Obolensky, A.; Beryozkin, A.; Tjon-Fo-Sang, M.J.H.; Schuil, J.; Klaver, C.C.W.; et al. Whole-exome sequencing identifies biallelic IDH3A variants as a cause of retinitis pigmentosa accompanied by pseudocoloboma. Ophthalmology. 2017, 124, 992–1003.]. Notably, all patients with autosomal recessive IDH3A-related RP and macular pseudocoloboma carried one null variant, which could account for the more severe phenotype observed in these individuals (lines 391-393).
Considering that Patient 1 is heterozygous for both the p.(Ala122Thr) and p.(Pro98Leu) variants, we propose that the IDH3A c.293C>T, p.(Pro98Leu) variant may represent a null mutation, contributing to the more severe symptoms observed in this patient, however second patient does not have a null variant, and also manifests pseudocoloboma, however less severe. Thus, in our report, one patient is homozygous for the p.(Ala122Thr) variant, which appears to retain some residual enzymatic activity and is associated with a milder phenotype, but manifests with pseudocoloboma. In contrast, the second patient, who harbors the p.(Pro98Leu) variant, presents with a markedly severe phenotype involving neurological manifestations. We hypothesize that p.(Pro98Leu) represents a null variant, supported by both the AlphaFold structural model and the clinical presentation. Accordingly, we revised the title to emphasize this potential genotype–phenotype correlation.
The author should include an AlphaFold based 3D-model of their identified variants to improve their discussion. Of course, experimentally it would be helpful if they could measures remaining enzymatic activity.
Authors’ response: AlphaFold based 3D-model was included in the manuscript (Figure 7). We currently have no means to experimentally measure the remaining enzymatic activity.
The scope of the journal are researchers with a general interest in molecular mechanisms. They are often no MD's and by far no ophthalmologists. Thus, the authors should take this into account and provide a revised version, which is more suited for a more general readership. E.g., provide more information on the identification of a pseudocoloboma in general and should use (more) arrows in the images to areas they discuss (See figure 1, how to identify the "well-defined" atrophic lesion" in the macula consistent with "macular pseudocoloboma").
Authors’ response: The authors thank you for your suggestion. The figures were modified as requested. For easier identification of pseudocoloboma, all figures now have arrows that point to the areas of pseudocoloboma.
Thank you for your consideration of this manuscript.
Sincerely,
Mirjana Bjeloš, Ana Ćurić, Benedict Rak, Biljana Kuzmanović Elabjer, Mladen Bušić, and Katja Rončević
Reviewer 2 Report
Comments and Suggestions for Authors
Disease-causing variants in the IDH3A gene have been linked to autosomal recessive retinitis pigmentosa 90 (RP90) and Leber congenital amaurosis. This study presents two patient cases: one compound heterozygous for IDH3A c.364G>A (p.Ala122Thr), and IDH3A c.293C>T (p.Pro98Leu); and another patient homozygous for the IDH3A c.364G>A (p.Ala122Thr) variant. This study provided evidence demonstrating the association between these specific variants and their clinical manifestations. This report also demonstrates that macular pseudocoloboma can occur even in the absence of a null variant. In contrast, more severe phenotypes, such as mitochondrial encephalopathy appear to be linked to the presence of a null allele. This is the first report describing the IDH3A c.293C>T (p.Pro98Leu) variant.
The discussion section is very thorough including functions of the IDH3 gene and each of its subunits and how they relate to diseases, individual mutations reported so far and their disease associations and mechanisms. The author also discussed the individual mutations and their reported diseases, and they also discussed macular pseudocoloboma in these patients.
Major issues:
- Some tests are performed but data is not reported, one example is ERG
- Case 2 is poorly written with lots of long and confusing sentences.
Minor issues:
- Mitochondrial encephalopathy was mentioned frequently in the manuscript but it is not known if these two patients have this condition.
- Please label the layers in OCT for Fig 3 and Fig 4, and use arrows to point to the defects
English for Case 2 description can be improved.
Author Response
Point-by-point response to reviewer
Dear Reviewer,
the authors thank you for your comments.
We have revised the manuscript entitled “Unraveling the IDH3A gene: macular pseudocoloboma in the absence of a null variant“ (manuscript ID: ijms-3544648) accordingly. The revised parts of the manuscript are highlighted in yellow.
Authors' answers to reviewer’s comments:
Reviewer 2:
Disease-causing variants in the IDH3A gene have been linked to autosomal recessive retinitis pigmentosa 90 (RP90) and Leber congenital amaurosis. This study presents two patient cases: one compound heterozygous for IDH3A c.364G>A (p.Ala122Thr), and IDH3A c.293C>T (p.Pro98Leu); and another patient homozygous for the IDH3A c.364G>A (p.Ala122Thr) variant. This study provided evidence demonstrating the association between these specific variants and their clinical manifestations. This report also demonstrates that macular pseudocoloboma can occur even in the absence of a null variant. In contrast, more severe phenotypes, such as mitochondrial encephalopathy appear to be linked to the presence of a null allele. This is the first report describing the IDH3A c.293C>T (p.Pro98Leu) variant.
The discussion section is very thorough including functions of the IDH3 gene and each of its subunits and how they relate to diseases, individual mutations reported so far and their disease associations and mechanisms. The author also discussed the individual mutations and their reported diseases, and they also discussed macular pseudocoloboma in these patients.
Major issues:
Some tests are performed but data is not reported, one example is ERG.
Authors’ response: Thank you for your comment. We appreciate the opportunity to clarify this point.
In this case, full-field electroretinography (FFERG) was performed following ISCEV standards (line 207). However, due to the advanced stage of retinal degeneration in inherited retinal dystrophies (IRDs) such as retinitis pigmentosa (RP), the ERG responses were extremely reduced and nearly undetectable. In such advanced IRD cases, particularly RP, it is not uncommon for scotopic and photopic responses to be nearly isoelectric. As a result, the amplitudes and implicit times of standard dark-adapted (DA) and light-adapted (LA) waveforms cannot be reliably measured or interpreted. Stating specific numerical values under these circumstances could be misleading, as the responses fall below the threshold of clinical significance and reproducibility. In clinical and research reporting of RP, it is common practice to describe the ERG as “extinguished,” “flat,” or “barely detectable,” especially when this aligns with the clinical phenotype. This qualitative reporting is widely accepted in the field when it accurately reflects a non-recordable response. Our description (lines 206-208) was intended to reflect this standard convention.
Case 2 is poorly written with lots of long and confusing sentences.
Authors’ response: Case 2 was rewritten as requested (lines 139-214).
Minor issues:
Mitochondrial encephalopathy was mentioned frequently in the manuscript but it is not known if these two patients have this condition.
Authors’ response: The first patient had infantile encephalopathy (lines 74-75, lines 416-417), the second patient had no neurological signs and symptoms (line 143).
Please label the layers in OCT for Fig 3 and Fig 4, and use arrows to point to the defects.
Authors’ response: The authors align with the reviewer's remark. Figures 3 and 4, as well as figure legends were supplemented as requested.
Sincerely,
Mirjana Bjeloš, Ana Ćurić, Benedict Rak, Biljana Kuzmanović Elabjer, Mladen Bušić, and Katja Rončević
Round 2
Reviewer 1 Report
Comments and Suggestions for Authors
The manuscript could now be published in the present form.